# Flexible versus Rigid Bronchoscopy for Tracheobronchial Foreign Body Removal in Children: A Comparative Systematic Review and Meta-Analysis

**DOI:** 10.3390/jcm13185652

**Published:** 2024-09-23

**Authors:** Alaa Safia, Uday Abd Elhadi, Rawnk Bader, Ashraf Khater, Marwan Karam, Taiser Bishara, Saqr Massoud, Shlomo Merchavy, Raed Farhat

**Affiliations:** 1Head & Neck Surgery Unit, Department of Otolaryngology, Rebecca Ziv Medical Center, Safed 1311001, Israel; udayabdi510@gmail.com (U.A.E.); dr.akhater@gmail.com (A.K.); marwan_ka@bezeqint.net (M.K.); taiser.bishara@gmail.com (T.B.); saqr.m@ziv.health.gov.il (S.M.); shlomo.m@ziv.health.gov.il (S.M.); raed.frhat88@gmail.com (R.F.); 2Research Wing, Safed 1311001, Israel; rawnaq1605@gmail.com

**Keywords:** tracheobronchial, foreign body, flexible, rigid, bronchoscopy

## Abstract

The removal of foreign bodies (FBs) from the airways of children is a critical procedure that can avert serious complications. While both flexible and rigid bronchoscopy techniques are employed for this purpose, their comparative efficacy and safety remain subjects of debate. Therefore, we conducted this investigation to compare between both procedures. Studies comparing flexible to rigid bronchoscopy (*n* = 14) were identified by searching PubMed, Scopus, Web of Science, Cochrane Library, and Google Scholar. We performed comparative meta-analyses of reported presentation characteristics and clinical outcomes, using fixed- and random-effects models. A diverse range of FB types and locations were identified. No difference was observed in the success rate of FB removal between flexible and rigid bronchoscopy (logOR = 0.27; 95%CI: −1.91:2.45). The rate of negative first bronchoscopy was higher in the flexible compared to the rigid group (logOR = 2.68; 95%CI: 1.68:3.67). Conversion rates to the alternative method were higher in the flexible bronchoscopy group. The overall complication rates were similar between both methods; however, the risk of desaturation was significantly lower with flexible bronchoscopy (logOR = −2.22; 95%CI: −3.36:−1.08). Flexible bronchoscopy was associated with a shorter length of hospital stay. The choice of bronchoscopy technique should be tailored to individual case characteristics.

## 1. Introduction

Foreign body aspiration or inhalation represents a significant cause of morbidity, and potentially mortality, among children worldwide. These incidents, resulting from the accidental inhalation of objects into the respiratory tract, necessitate prompt and effective medical intervention to prevent severe complications, including airway obstruction, recurrent pneumonia, and chronic cough [1]. The standard approach to managing these cases involves the use of bronchoscopy, a procedure that allows for the visualization and removal of foreign bodies from the airway [2,3].

Bronchoscopy can be performed using either a rigid or a flexible bronchoscope. Rigid bronchoscopy has been traditionally favored in pediatric cases due to its dual role in airway management and foreign body retrieval, offering robust control and the ability to use larger instruments [4]. However, advancements in technology and technique have seen an increase in the use of flexible bronchoscopy [5]. This method is perceived as less invasive, with a potentially lower risk of complications and the ability to access distal airways, thus broadening the diagnostic and therapeutic scope of the procedure [5,6,7].

Despite these advancements, the choice between rigid and flexible bronchoscopy for foreign body removal in children remains a subject of debate. Proponents of rigid bronchoscopy highlight its efficacy and safety profile, supported by decades of clinical experience. Meanwhile, advocates for flexible bronchoscopy emphasize its minimal invasiveness, patient comfort, and versatility [8]. This divergence in practice preferences underscores the need for a comprehensive comparison of the two methods, focusing on their efficacy and safety outcomes in the pediatric population.

Given the lack of conclusive evidence in this regard, a systematic review and meta-analysis of the available evidence on flexible versus rigid bronchoscopy in the removal of foreign bodies among children can provide valuable insights into optimizing the management of these potentially life-threatening incidents. This study aimed to compare the efficacy and safety of flexible and rigid bronchoscopy for the removal of foreign bodies in children, drawing on the latest evidence to guide clinical practice. By analyzing outcomes related to efficacy and safety endpoints, this research seeks to elucidate the optimal bronchoscopic approach in pediatric cases of foreign body aspiration or inhalation.

## 2. Materials and Methods

### 2.1. Study Design & Protocol

This systematic review and meta-analysis were designed to compare the efficacy and safety of flexible versus rigid bronchoscopy for the removal of foreign bodies in the pediatric population. The study protocol was developed following the Preferred Reporting Items for Systematic Reviews and Meta-Analyses (PRISMA) guidelines [9]. The study protocol was previously registered on PROSPERO (CRD42024519719). Approval from the Institutional Review Board was waived due to the non-involvement of human participants in this research.

### 2.2. Eligibility Criteria (PICOS Framework)

The eligibility criteria were designed using the PICOS (Population, Intervention, Comparison, Outcomes, and Study Design) framework [10]. Inclusion criteria included all the following points combined: (1) original research papers with either randomized or non-randomized comparative studies (study design), (2) children (<18 years of age) with suspected tracheobronchial foreign body (population), (3) comparing flexible (intervention) to rigid bronchoscopy (comparison), and (4) reporting any clinical outcomes (outcomes, listed below). Meanwhile, the exclusion criteria were as follows: (1) non-original and secondary research (i.e., reviews, editorials, etc.), (2) non-comparative studies, (3) adult patients, (4) lack of relevant outcomes, or (5) duplicates or overlapping datasets.

### 2.3. Literature Search

A systematic literature search was conducted across multiple electronic databases, including PubMed, Scopus, Cochrane Library, and Web of Science, from their inception to 1 March 2024. We also searched the grey literature (the first 200 records of Google Scholar), as per guidelines [11]. The search strategy combined terms related to rigid and flexible bronchoscopy, foreign bodies, and children, without language restrictions (Appendix A). Additional studies were identified through manually searching references of included studies and relevant reviews in addition to the “similar articles” function on PubMed [12].

### 2.4. Article Screening

The screening process was conducted following the removal of duplicated citations through EndNote software (Version X8). Two reviewers independently screened the titles and abstracts of identified records for eligibility. Full-text articles were then reviewed to confirm inclusion based on the predefined criteria. Disagreements between reviewers were resolved through discussion or consultation with a third reviewer.

### 2.5. Data Extraction

Data were extracted independently by two reviewers using a standardized data extraction form. Extracted information included study characteristics (author, year, country, design, sample size), participant demographics (age, gender), details of the foreign body (type, location), bronchoscopy method specifics (flexible or rigid, assistive instrument, and indication), and detailed outcome data. Discrepancies were resolved by consensus or third-party adjudication. Outcome measures included both efficacy and safety endpoints. Efficacy endpoints included (1) successful removal (without the need for subsequent procedures), (2) failed extraction, necessitating alternative intervention, (3) negative first bronchoscopy rate: the proportion of initial bronchoscopies that did not retrieve a foreign body, necessitating a repeat bronchoscopy, (4) rate of repeated bronchoscopy (of same method), (5) conversion rate to the other method, (6) operative time, and (7) length of hospital stay (LOS). The safety endpoints included complications (overall, major, and type), death, and post-bronchoscopy intubation and intensive care unit (ICU) admission.

### 2.6. Methodological Quality Assessment

The quality of included studies was assessed using the Newcastle Ottawa Scale (NOS) tool for assessing the quality of cohort studies. Each study was given an overall rating of good, fair, or poor quality according to the number of stars given to each of the assessed domains (selection “total of 4 stars”, comparability “total of 2 stars”, and outcome/exposure “total of 3 stars”).

### 2.7. Data Analysis

All analyses were performed as per protocol. One study was published in Korean, and it was translated to English with the help of an artificial intelligence tool. The translation was later validated by a native Korean colleague. Notably, data transformation (from median and interquartile range to mean and standard deviation) was performed in some studies [5,6,13] to standardize the reported effect measure using validated transformation formulas [14]. Additionally, some outcomes (i.e., suboptimal FB extraction [6], extraction in the operating room [15]) were reported in only one study; thus, a meta-analysis was not performed.

Meta-analyses were performed to calculate pooled effect sizes, expressed as mean differences (MD) for continuous outcomes and log odds ratios (logOR) for binary outcomes, both with 95% confidence intervals (CIs) [16]. Additionally, to calculate the pooled rate in each bronchoscopy method, separately, a meta-analysis of proportion was done using the metaprop command [17]. Subgroup analyses were only performed for FB location and nature, assistive extraction instruments, and complication type.

The statistical model, either random-effects or fixed-effects, was chosen based on observed statistical heterogeneity. When heterogeneity was high, the random-effects model was selected, and when it was insignificant, the fixed-effects model was the model of choice. Significant heterogeneity was defined as I^2^ above 50% with a *p* value < 0.05 [18]. Leave-one-out sensitivity analyses were conducted to explore the impact of individual studies on the overall results; however, in analyses reported by only two studies, the sensitivity analysis was not feasible. Publication bias was assessed through funnel plot inspection and Egger’s regression test [19]. Statistical analyses were conducted using STATA (version 18).

## 3. Results

### 3.1. Literature Review and Screening Results

The literature search yielded 659 citations, of which 261 duplicates were identified and removed through EndNote. Of the 398 screened articles, 26 studies were eligible for full-text screening (Figure 1). Of them, 14 studies were excluded (single-group studies accounted for 13 and mixed-age population accounted for one study). The manual search resulted in two additional articles, resulting in a total number of 14 studies eligible for data synthesis and analysis [5,6,7,13,15,20,21,22,23,24,25,26,27,28].

### 3.2. Characteristics of Included Studies

All of the included studies were retrospective cohort studies, each of which was conducted in a different country (14 in total). They involved a total of 9546 patients suspected of FB inhalation or aspiration, of which 3968 received rigid bronchoscopy and 5578 received flexible bronchoscopy. Patients’ ages and genders are described in Table 1, however, only a few studies stratified baseline demographics based on the bronchoscopy method. The indication for bronchoscopy was merely diagnostic in six studies [6,13,20,21,23,25], combined diagnostic/therapeutic in two studies [5,15], and non-defined clearly in the remaining studies. The model of each bronchoscopy method was clearly defined in only five studies [5,6,15,21,25].

### 3.3. Methodological Quality of Included Studies

Following the NOS’s guidelines in assessing the quality of included studies, only three studies were deemed of good quality (Table 2) [7,13,21]. All of the remaining studies had poor quality. This poor quality was mainly attributed to the lack of confounder control either regarding patients’ demographic (i.e., age, gender) or non-demographic characteristics (i.e., indication, FB type and location).

### 3.4. Presentation Patterns

The location and nature of extracted FBs, as well as the instruments used for extraction, are reported in Table 3. In summary, there was no significant difference between the flexible and rigid bronchoscopes in terms of FB location, nature, or used instruments. Regarding the location of extracted FBs, four main sites were reported: main/lobar bronchi, trachea, and larynx. FBs were either organic or inorganic. The most commonly used assisting instruments were baskets and forceps, and, less frequently, crocodile and tweezers. The mean inhalation to bronchoscopy time was similar between both groups [MD = −1.22 days; 95%CI: −9.68:7.25].

### 3.5. Efficacy Endpoints

Successful FB Removal

No significant differences in success were observed between both rigid and flexible bronchoscopy [7 studies, logOR = 0.27; 95%CI: −1.91:2.45, I^2^ = 89.81%, *p* = 0.001] (Figure 2). The sensitivity analysis showed no change in the reported estimate (Appendix A).

2.Failure

No significant difference in failure rate was noted between rigid and flexible bronchoscopy [6 studies, logOR = 0.32; 95%CI: −1.43:2.07, I^2^ = 83.03%, *p* = 0.001] (Figure 3). The sensitivity analysis showed no change in the reported estimate (Appendix A).

3.Negative First Bronchoscopy Rate

Three studies reported the rate of negative first bronchoscopy. This rate was substantially high in the flexible bronchoscopy group, accounting for 66% [95%CI: 40–92%] of suspected cases. Meanwhile, this rate was 9% [95%CI: 0–22%] in the rigid bronchoscopy group. The comparative meta-analysis showed a significantly higher risk of negative first bronchoscopy in the flexible group as compared to the rigid group [logOR = 2.68; 95%CI: 1.68:3.67] (Figure 4). This finding was supported by the lack of heterogeneity [I^2^ = 0%, *p* = 0.64].

4.Repeated Bronchoscopy Rate

Repeated bronchoscopy of the same method was investigated by three studies, all of which reported no occurrences in the flexible group, compared to 2.24% in the rigid group. This is supported by the meta-analysis, which showed no significant difference in the risk of repeated bronchoscopy between both methods [logOR = −1.37; 95%CI: −3.25:0.51] (Appendix A). No heterogeneity was observed [I^2^ = 0%, *p* = 0.69].

5.Bronchoscopy Conversion Rate

Four studies reported the rate of conversion to the other method of bronchoscopy. In the rigid group, 6% [95%CI: 3–8%] of cases converted to flexible bronchoscopy, while 27% [95%CI: 13–41%] of cases in the flexible group converted to rigid bronchoscopy. The comparative meta-analysis showed a significant increase in the risk of conversion in favor of the flexible bronchoscopy group compared to the rigid one [logOR = 1.54; 95%CI: 0.03–3.05, I^2^ = 78.33%, *p* = 0.001] (Figure 5). The sensitivity analysis revealed different effect estimates (Appendix A). This reflects the low accuracy of this finding.

6.Operative time

The mean operative time following bronchoscopy was reported in four studies. The pooled meta-analysis showed no significant difference in the operative time between flexible and rigid bronchoscopy [MD = −1.98; 95%CI: −9.60:5.65, I^2^ = 81.30%, *p* = 0.001] (Figure 6). The sensitivity analysis showed no change in the reported estimate (Appendix A).

7.Length of hospital stay

The length of hospital stay was reported in four studies. The pooled meta-analysis revealed that flexible bronchoscopy was associated with a significantly lower length of hospital stay than rigid bronchoscopy, by 4.8 h [MD = −0.20 days; 95%CI: −0.28:−0.12]. No heterogeneity was observed [I^2^ = 11.19%, *p* = 0.32] (Figure 7).

### 3.6. Safety Endpoints

Complications

The overall complication rate of both rigid and flexible bronchoscopy was reported in 12 studies, and the pooled meta-analysis showed no significant difference in the risk of complications between both groups [logOR = −0.07, 95%CI: −0.26:0.11]. This is supported by the lack of significant heterogeneity [I^2^ = 34.68%, *p* = 0.11] (Figure 8). Although the Funnel plot showed a deviation towards the left side of the plot (Appendix A), Egger’s regression test revealed no significant risk of publication bias [*p* = 0.23].

The following complications were reported from highest to lowest: bleeding, laryngeal edema, fever, bronchospasm, laryngospasm, emphysema, pneumothorax, desaturation, transient hypoxia, and cough (Table 4). Similar complications were observed between flexible and rigid bronchoscopies; however, the risk of desaturation was significantly reduced in the flexible method [logOR = −2.22; 95%CI: −3.36:−1.08, I^2^ = 77.65%, *p* = 0.03].

2.Major Complications

Major complications following bronchoscopy occurred only in four studies, of which 0% of cases in the flexible bronchoscopy group had major complications, while 1.4% of cases in the rigid bronchoscopy group experienced them. The meta-analysis, however, showed no significant difference in the risk of major complications between both methods [logOR = −1.64; 95%CI: −3.42:0.15, I^2^ = 89.02%, *p* = 0.001] (Figure 9).

3.Death/Mortality

Death following bronchoscopy was investigated by four studies, all of which reported no occurrences of death following bronchoscopy. This is supported by the meta-analysis, which showed no significant difference in the risk of death between both methods [logOR = −0.68; 95%CI: −2.68:1.32] (Figure 10). No heterogeneity was observed [I^2^ = 0%, *p* = 0.63].

4.Post-bronchoscopy Intubation

The need for intubation post-bronchoscopy was investigated by three studies, all of which reported no intubations in those who underwent flexible bronchoscopy, while 5.8% of those who underwent rigid bronchoscopy needed intubation. The pooled meta-analysis, however, revealed no significant difference in the risk of intubation between both methods [logOR = −1.02; 95%CI: −2.83:0.78] (Appendix A). No heterogeneity was observed [I^2^ = 0%, *p* = 0.50].

5.Post-bronchoscopy pediatric ICU admission

The need for repeated ICU admission post-bronchoscopy was investigated by only two studies. The pooled meta-analysis revealed no significant difference in the risk of ICU admission post-bronchoscopy between both methods [logOR = 1.03; 95%CI: −0.34:2.39] (Appendix A). No heterogeneity was observed [I^2^ = 0%, *p* = 0.40].

## 4. Discussion

Our findings suggested that both flexible and rigid bronchoscopy were comparably effective in the successful extraction of foreign bodies, with no significant difference in the likelihood of success or failure. Moreover, both flexible and rigid bronchoscopy demonstrated similar safety profiles with no significant difference in overall complication rates. However, certain nuances in our results merit a detailed discussion.

### 4.1. Presentation Patterns and Efficacy Endpoints

Our analysis did not reveal significant differences in the location or nature of extracted foreign bodies, or the instruments used, between flexible and rigid bronchoscopy. This suggests a broad equivalence in the technical capabilities of both methods for addressing a wide range of foreign body cases in pediatric patients. The lack of significant difference in the mean inhalation to bronchoscopy time further supports this equivalence.

Notably, the pooled meta-analysis demonstrated no significant difference in the likelihood of successful foreign body removal between the two methods, despite considerable statistical heterogeneity. This heterogeneity might reflect variability in study populations, types of foreign bodies encountered, or operator expertise, indicating that the choice between flexible and rigid bronchoscopy could be influenced by these factors rather than inherent differences in efficacy [5,7,21,25]. The variability in study populations, types of foreign bodies, and operator expertise observed in the included studies introduces significant challenges in directly comparing the efficacy of flexible and rigid bronchoscopy. These factors likely influence procedural success and safety outcomes, and their impact may obscure any inherent differences between the two techniques. Consequently, the comparability of flexible and rigid bronchoscopy must be interpreted cautiously, as the choice of technique may be influenced by the specific context of each procedure, rather than by objective efficacy differences. Future studies should seek to minimize this variability by focusing on more homogeneous populations, or by providing stratified analyses that account for operator expertise, foreign body characteristics, and other key variables.

Risk factors for procedural failure or the need for multiple attempts at bronchoscopy might include severe airway inflammation [29], previous unsuccessful retrieval attempts [30], and certain FB characteristics (e.g., vegetative or sharp objects) [31]. Recognizing these predictors and risk factors can aid in pre-procedural planning, including the selection of the bronchoscopy method, preparation for potential difficulties, and counseling of patients and families regarding procedural risks and expectations. Given the scarcity of data regarding these variables, a quantitative assessment of the impact of these factors in determining examined outcomes was not possible. Therefore, future research should address these points and stratify their outcome data based on these factors.

### 4.2. Negative First Bronchoscopy and Conversion Rates

The finding that flexible bronchoscopy is associated with a higher rate of missed foreign bodies, particularly in the lower airways, is indeed unexpected and may not be due to the limitations of the flexible bronchoscope itself. Flexible bronchoscopes are generally better suited for inspecting the subsegmental bronchi, especially in small children, due to their size and flexibility [32]. Conversely, the rigid bronchoscope’s larger diameter and direct line of sight might offer superior visualization and mechanical advantage for FB retrieval, especially in proximal airways [33].

It is possible that the higher rate of missed foreign bodies is influenced by the experience of the bronchologist, or the quality of the video equipment used (e.g., monitor vs. direct visual inspection). Unfortunately, given the lack of data in this regard, we could not analyze the impact of these factors on the outcome. Further research is needed to explore how these factors impact the effectiveness of flexible bronchoscopy in foreign body detection.

The substantial proportion of cases requiring conversion from flexible to rigid bronchoscopy underscores a critical aspect of bronchoscopic FB removal: the need for individualized approach selection based on patient anatomy, FB characteristics, and the presenting scenario. The decision to convert may also reflect the operator’s threshold for switching methods when faced with procedural difficulties, such as inability to retrieve the FB, poor visualization, or patient instability [34]. This suggests that in some cases, starting with a rigid bronchoscope might be more pragmatic, especially in scenarios historically associated with higher conversion rates, such as large or irregularly shaped FBs lodged in the proximal airways. These findings highlight the importance of having both options available, emphasizing the complementary nature of the two techniques, rather than a competition between them.

In cases where conversion from flexible to rigid bronchoscopy is required, there is a plausible risk that the preceding flexible bronchoscopy could exacerbate airway irritation or worsen the foreign body’s position, making removal more challenging. Unfortunately, the included studies did not provide specific data on operative times or complication rates in these converted cases. This limitation highlights the potential for increased difficulty in foreign body removal following conversion, further supporting the argument for initial use of rigid bronchoscopy in cases where the foreign body is expected to be difficult to retrieve, or in cases where flexible bronchoscopy has a higher likelihood of failure.

Predictors for conversion could include the size and nature of the FB, its anatomical location, and the patient’s age or airway anatomy [35]. Identifying these predictors through future research could aid in developing guidelines or algorithms to assist clinicians in selecting the most appropriate initial bronchoscopic approach, potentially reducing the need for conversions and associated procedural risks.

### 4.3. Operative Time and Length of Hospital Stay

The analysis revealed no significant difference in operative time between the two methods, suggesting that the choice of bronchoscopy technique does not necessarily impact procedural duration. However, the significant reduction in length of hospital stay associated with flexible bronchoscopy, even though modest, suggests potential advantages in patient recovery and healthcare resource utilization. This could reflect fewer complications, faster recovery from anesthesia, or other factors associated with the less invasive nature of flexible bronchoscopy.

### 4.4. Safety Endpoints

The overall similarity in complication rates between flexible and rigid bronchoscopy is reassuring, indicating that both methods are safe for pediatric FB removal when performed by experienced operators. However, the specific reduction in the risk of desaturation with flexible bronchoscopy is a significant finding that highlights the method’s potentially less invasive nature. Desaturation during bronchoscopy can be a marker of airway compromise, procedural difficulty, or prolonged procedure time, suggesting that flexible bronchoscopy might offer advantages in maintaining patient stability and oxygenation [36].

Despite these findings, the high heterogeneity observed in our analysis of safety endpoints, especially concerning major complications and desaturation, indicates variability in how complications are reported, managed, and prevented across different studies and clinical settings. This underscores the need for standardized reporting of bronchoscopy-related complications.

### 4.5. Limitations and Future Directions

While our findings contribute to the body of evidence comparing flexible and rigid bronchoscopy, the considerable heterogeneity observed in several analyses highlights the variability in study designs, populations, and procedural techniques, particularly operator experience. Future research should aim to standardize reporting and procedural protocols to reduce this variability. Despite the impressive number of reviewed studies and the large total number of patients included in our meta-analysis, it is important to recognize the issue of underreporting, particularly from larger pediatric hospitals. Many such institutions may not publish their data on foreign body removal, which could lead to a significant reporting bias. This underreporting may influence the generalizability of our findings and obscure the true rates of success and complications associated with both flexible and rigid bronchoscopy.

Although our analysis did not reveal significant differences between flexible and rigid bronchoscopy regarding the location or nature of foreign bodies, several important factors were not accounted for due to limitations in the available data. Variables such as the size and sharpness of the foreign body, its relative size to the airway lumen, whether the object is fixed or mobile, and more peripheral locations such as subsegmental airways may significantly influence both procedural success and the choice of bronchoscopy method. These factors are crucial, and should be considered in clinical decision-making, and future studies should aim to stratify outcomes based on these parameters to provide more nuanced recommendations.

Additionally, the high conversion rate from flexible to rigid bronchoscopy suggests a need for further investigation into the specific scenarios or characteristics that predict the need for conversion, which could guide initial method selection and pre-procedural planning.

## 5. Conclusions

While both flexible and rigid bronchoscopy were found to be effective and safe for foreign body removal in pediatric patients based on the available data, the significant heterogeneity observed across the included studies suggests caution in generalizing these findings. The variability in study designs, patient characteristics, and procedural techniques impacts the comparability of the two methods. Therefore, individualized decisions based on the clinical context, patient factors, and operator expertise remain crucial, and further research is needed to refine these conclusions in more homogeneous study settings.

## Figures and Tables

**Figure 1 jcm-13-05652-f001:**
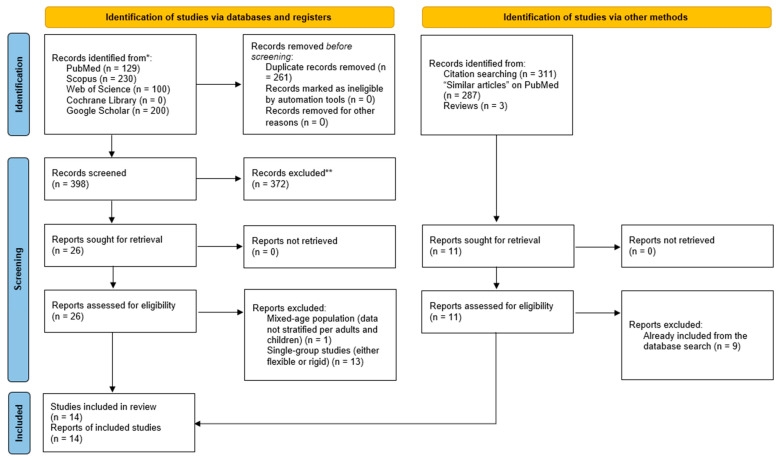
PRISMA diagram showing the results of the literature search and screening processes. * records identified during the initial database search before duplicate identification and removal; ** excluded records during the title/abstract screening phase.

**Figure 2 jcm-13-05652-f002:**
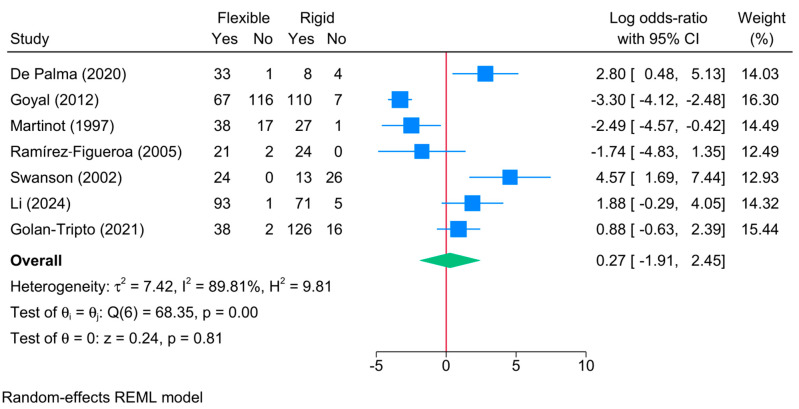
Forest plot showing the difference in successful foreign body extraction between flexible and rigid bronchoscopy [5,6,7,21,22,25,27].

**Figure 3 jcm-13-05652-f003:**
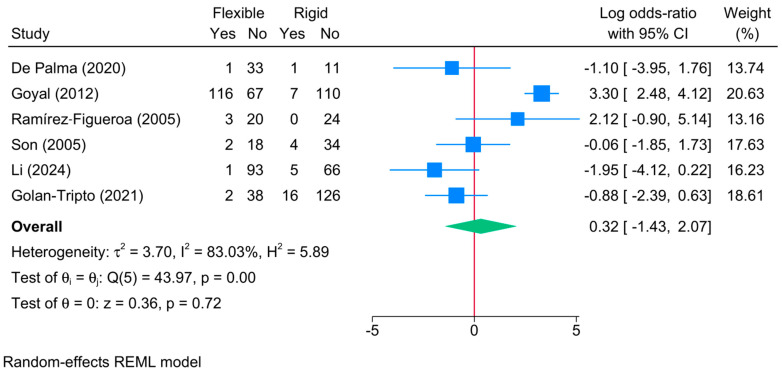
Forest plot showing the difference in failed foreign body extraction between flexible and rigid bronchoscopy [5,6,7,21,22,26].

**Figure 4 jcm-13-05652-f004:**
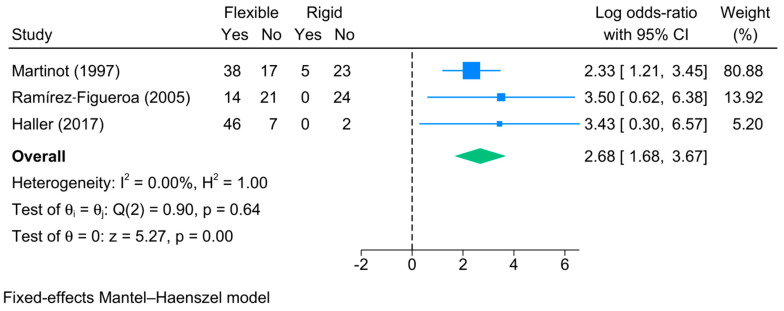
Forest plot showing the difference in negative first bronchoscopy between flexible and rigid bronchoscopy [5,23,25].

**Figure 5 jcm-13-05652-f005:**
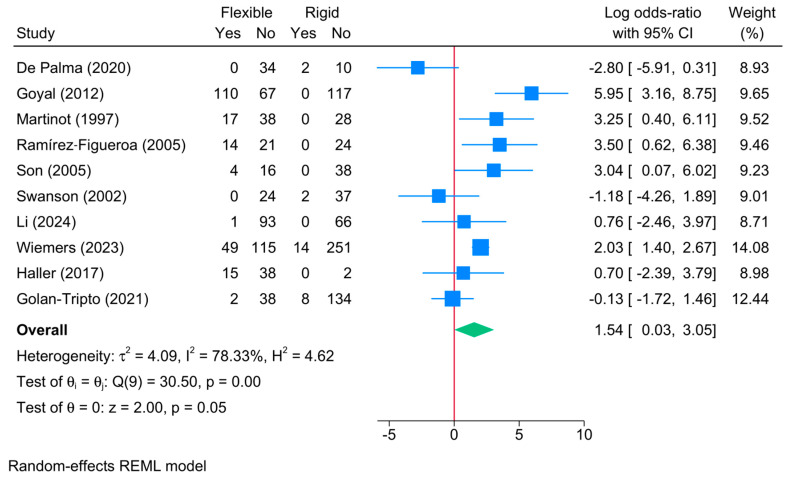
Forest plot showing the difference in conversion between flexible and rigid bronchoscopy [5,6,7,21,22,23,25,26,27,28].

**Figure 6 jcm-13-05652-f006:**
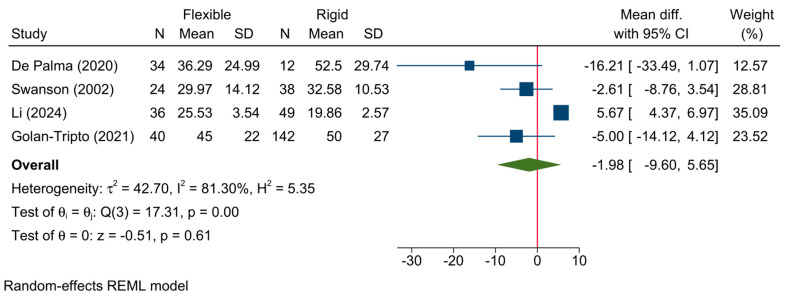
Forest plot showing the difference in operative time between flexible and rigid bronchoscopy [6,7,21,27].

**Figure 7 jcm-13-05652-f007:**
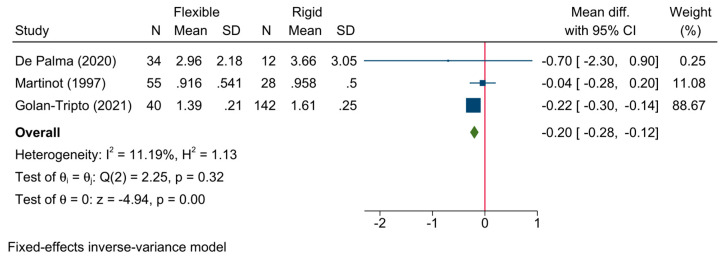
Forest plot showing the difference in length of hospital stay between flexible and rigid bronchoscopy [6,21,25].

**Figure 8 jcm-13-05652-f008:**
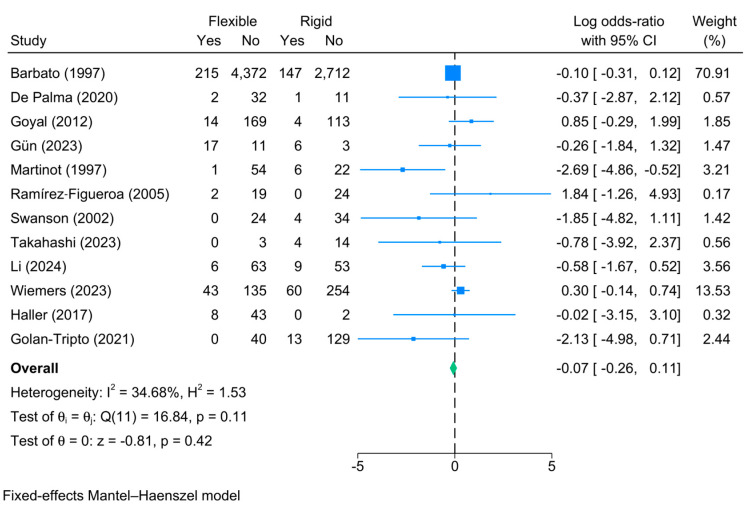
Forest plot showing the difference in total complications between flexible and rigid bronchoscopy [5,6,7,13,15,20,21,22,23,25,27,28].

**Figure 9 jcm-13-05652-f009:**
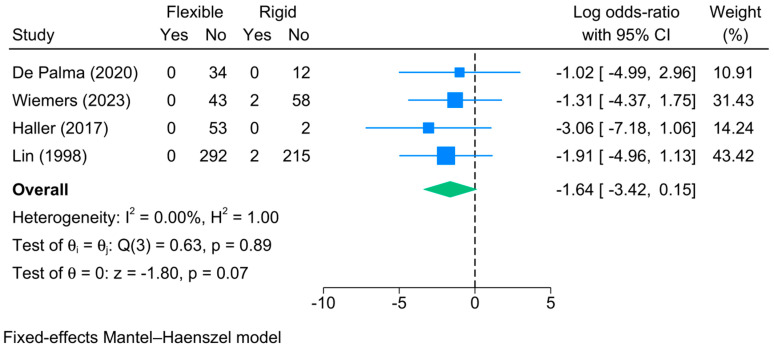
Forest plot showing the difference in major complications between flexible and rigid bronchoscopy [6,23,24,28].

**Figure 10 jcm-13-05652-f010:**
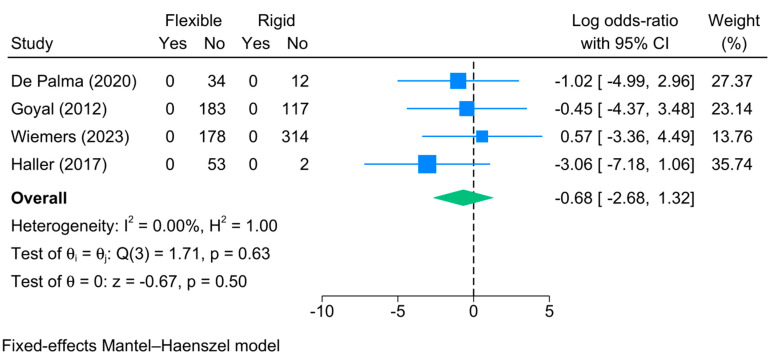
Forest plot showing the difference in death between flexible and rigid bronchoscopy [6,22,23,28].

**Table 1 jcm-13-05652-t001:** The characteristics of studies comparing flexible to rigid bronchoscopy in children with foreign body aspiration or inhalation.

Author	Country	Design	Sample		Age (year)		Gender (M:F)		Indication of Bronchoscopy		Bronchoscopy Model	
			RB	FB	RB	FB	RB	FB	RB	FB	RB	FB
Barbato (1997) [20]	Austria	RC	2859	4587	Children (no defined age group)		-	-	Diagnostic (whole population)		NCD	
De Palma (2020) [6]	Italy	RC	12	34	0.5–1.87 *	0.54–4.87 *	M:F = 33:18		Diagnostic (whole population)		Storz 10338 CD, 10339 BB, 10339 C	Pentax FB 8V, Storz 11003BC3
Goyal (2012) [22]	India	RC	183	135	<12	<12	M:F = 126:88		NCD		NCD	
Gün (2023) [15]	Turkey	RC	9	28	0.29–5.25 *		M:F = 20:17		Diagnostic = 23; Therapeutic = 14		Karl Storz Endoscope, Tuttlingen	Fiberoptic Karl Storz Endoscope
Martinot (1997) [25]	France	RC	28	55	1		M:F = 57:26		Diagnostic = 28; Therapeutic = 55		Storz equipment	Pentax or Olympus bronchoscope
Ramírez-Figueroa (2005) [5]	Mexico	RC	24	35	0.37–16 *		M:F = 35:24		Diagnostic = 12; Therapeutic = 47		-	Pentax
Son (2005) [26]	South Korea	RC	42	20	<15		M:F = 47:17		NCD		NCD	
Swanson (2002) [27]	USA	RC	39	24	0.37–7.62 *		M:F = 28:11		NCD		NCD	
Takahashi (2023) [13]	Japan	RC	18	3	0–10 *		M:F = 15:3	M:F = 2:1	Diagnostic = 18; Therapeutic = 3		NCD	
Li (2024) [7]	China	RC	79	94	1–8 *		M:F = 54:25	M:F = 65:29	NCD		NCD	
Wiemers (2023) [28]	Germany	RC	314	178	2.97 (2.78)		M:F = 277:152		NCD		NCD	
Haller (2017) [23]	Switzerland	RC	2	53	0.89 (0.58–1.48) *		M:F = 25:45		Diagnostic (whole population)		NCD	
Golan-Tripto (2021) [21]	Israel	RC	142	40	2 (1.3–8) *	2 (1.6–8.6) *	M:F = 84:58	M:F = 22:18	Diagnostic (whole population)		Hopkins bronschoscope (Storz, 2.9 mm, 37 cm)	Olympus, external diameter of 3.6 mm, working channel 1.2 mm
Lin (1998) [24]	Taiwan	RC	217	292	Children (no defined age group)		-	-	NCD		NCD	

* Indicates that the data are reported as either range or median (range). RC: retrospective cohort; M:F: the number of males to females (not ratio); NCD: not clearly defined; RB: rigid bronchoscopy; FB: flexible bronchoscopy.

**Table 2 jcm-13-05652-t002:** The methodological quality of included studies assessed by the Newcastle Ottawa Scale for cohort studies.

Author (YOP)	Selection			Comparability		Outcome			Total Grade
	Representativeness	Selection of Non-Exposed Cohort	Ascertainment of Exposure	Control for Age, Sex, and Marital Status	Control for Other Factors	Assessment of Outcome	Follow-Up Long Enough for Outcomes to Occur	Adequacy of Follow-Up	
Barbato (1997) [20]	☆	☆	☆	-	-	☆	☆	☆	Poor
De Palma (2020) [6]	☆	☆	☆	-	-	☆	☆	☆	Poor
Goyal (2012) [22]	☆	☆	☆	-	-	☆	☆	☆	Poor
Gün (2023) [15]	☆	☆	☆	-	-	☆	☆	☆	Poor
Martinot (1997) [25]	☆	☆	☆	-	-	☆	☆	☆	Poor
Ramírez-Figueroa (2005) [5]	☆	☆	☆	-	-	☆	☆	☆	Poor
Son (2005) [26]	☆	☆	☆	-	-	☆	☆	☆	Poor
Swanson (2002) [27]	☆	☆	☆	-	-	☆	☆	☆	Poor
Takahashi (2023) [13]	☆	☆	☆	☆	-	☆	☆	☆	Good
Li (2024) [7]	☆	☆	☆	☆	-	☆	☆	☆	Good
Wiemers (2023) [28]	☆	☆	☆	-	-	☆	☆	☆	Poor
Haller (2017) [23]	☆	☆	☆	-	-	☆	☆	☆	Poor
Golan-Tripto (2021) [21]	☆	☆	☆	☆	-	☆	☆	☆	Good
Lin (1998) [24]	☆	☆	☆	-	-	☆	☆	☆	Poor

YOP: year of publication. ☆ indicates proper methodology in each respective part of the assessment tool.

**Table 3 jcm-13-05652-t003:** A summary of the meta-analysis difference between rigid and flexible bronchoscopy, regarding foreign body’s location, nature, and assistive extraction instruments.

Variable	Category	K	LogOR	95%CI	*p*
**Location**	Larynx	1	−0.69	[−3.67, 2.30]	0.35
	Trachea	3	0.09	[−2.48, 2.67]	
	Main Bronchus	3	0.11	[−0.22, 0.43]	
	Lobar Bronchus	3	−0.78	[−1.92, 0.36]	
**Nature**	Organic	4	−0.11	[−0.41, 0.19]	0.11
	Inorganic	4	0.53	[−0.21, 1.28]	
**Instrument**	Basket	2	0.00	[−0.96, 0.96]	0.62
	Forceps	2	−1.30	[−2.93, 0.33]	
	Combined *	1	0.51	[−2.60, 3.62]	
	Crocodile	1	−0.12	[−2.34, 2.10]	
	Tweezers	1	1.26	[−2.68, 5.19]	
**Inhalation-to-bronchoscopy time**		2	−1.22	[−9.68, 7.25]	-

* This data refers to the combined use of both basket and forceps during bronchoscopic removal of foreign bodies. K: number of studies per variable; logOR: log odds ratio; CI: confidence interval; *p*: indicates the significance level of effect modification by analyzed subgroups.

**Table 4 jcm-13-05652-t004:** A summary of the meta-analysis findings of the difference in complication types between flexible and rigid bronchoscopy.

Complication	K	LogOR	95%CI	I^2^	*p*
Bleeding	6	−0.18	[−0.57, 0.20]	0%	0.84
Laryngeal Edema	5	−0.70	[−1.51, 0.12]	62.19%	0.03
Fever	4	−0.21	[−0.72, 0.31]	0%	0.79
Bronchospasm	3	−0.09	[−0.45, 0.27]	33.31%	0.22
Laryngospasm	3	−0.13	[−2.53, 2.28]	13.56%	0.28
Transient Hypoxia	2	−0.66	[−2.01, 0.69]	0%	0.68
Desaturation	2	−2.22	[−3.36, −1.08]	77.65%	0.03
Emphysema	2	−0.34	[−1.69, 1.00]	63.67%	0.10
Cough	2	0.39	[−0.42, 1.20]	0%	0.84
Pneumothorax	2	1.47	[−1.19, 4.12]	0%	0.53

K: number of studies per variable; logOR: log odds ratio; CI: confidence interval; I^2^: a measure of statistical heterogeneity; *p*: indicates the significance level of reported heterogeneity (I^2^).

## Data Availability

The analyzed dataset in this research study can be provided by the corresponding author upon reasonable request.

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
