# Peer review of "Flexible versus Rigid Bronchoscopy for Tracheobronchial Foreign Body Removal in Children: A Comparative Systematic Review and Meta-Analysis"

_jcm, 2024, doi:10.3390/jcm13185652_

Round 1
Reviewer 1 Report
Comments and Suggestions for Authors
Congratulations on this well written meta-analysis paper. In only have one comment: it is to be assumed, that in cases, where a conversion from a flexible to rigid bronchoscopy was needed, the removal of the foreign body with the rigid brochoscopy is more difficult because the preceding flexible bronchoscopy has worsened the situation. Do you have any analysis regarding the time and complication rate of the rigid bronchoscopy in these converted cases? This, besides the high conversion rate, is another argument in favor of rigid bronchoscopy for foreign body removal.
Author Response
Comment 1: Congratulations on this well written meta-analysis paper. In only have one comment: it is to be assumed, that in cases, where a conversion from a flexible to rigid bronchoscopy was needed, the removal of the foreign body with the rigid brochoscopy is more difficult because the preceding flexible bronchoscopy has worsened the situation. Do you have any analysis regarding the time and complication rate of the rigid bronchoscopy in these converted cases? This, besides the high conversion rate, is another argument in favor of rigid bronchoscopy for foreign body removal.
Response: Thank you for your valuable comment. We agree that conversions from flexible to rigid bronchoscopy may increase procedural difficulty, possibly due to airway irritation or worsening of the foreign body’s position. However, the studies included in our meta-analysis did not report sufficient detailed data on operative times and complication rates specifically for cases that required conversion from flexible to rigid bronchoscopy. While our results suggest a high conversion rate, we were unable to conduct a specific analysis on these converted cases due to the limitations of the available data. We have now added a statement highlighting this limitation and the potential impact of conversion on the outcomes of rigid bronchoscopy: "In cases where conversion from flexible to rigid bronchoscopy is required, there is a plausible risk that the preceding flexible bronchoscopy could exacerbate airway irritation or worsen the foreign body’s position, making removal more challenging. Unfortunately, the included studies did not provide specific data on operative times or complication rates in these converted cases. This limitation highlights the potential for increased difficulty in foreign body removal following conversion, further supporting the argument for initial use of rigid bronchoscopy in cases where the foreign body is expected to be difficult to retrieve or in cases where flexible bronchoscopy has a higher likelihood of failure."
Reviewer 2 Report
Comments and Suggestions for Authors
Please see the attached file.

Author Response
Comment 1. Line 304: The analysis did not reveal significant differences in the location or nature of extracted foreign bodies or the instruments used between flexible and rigid bronchoscopy. Important factors that could significantly impact outcomes or influence the choice of procedure, such as the size of the foreign body, its sharpness, the relative size of the foreign body to the airway lumen where it is lodged, whether the object is fixed or mobile, and more peripheral locations like subsegmental airways, were not considered or mentioned in the article.
Response: Thank you for your insightful comment. We acknowledge the importance of additional factors such as the size, sharpness, mobility, and location of the foreign body (e.g., subsegmental airways) in influencing procedural outcomes and the choice between flexible and rigid bronchoscopy. While our analysis did not have sufficient data to perform subgroup analyses based on these characteristics, we agree that these are critical considerations. We have revised the manuscript to mention these factors as potential limitations in the interpretation of our findings and have emphasized the need for future research to explore these variables: "Although our analysis did not reveal significant differences between flexible and rigid bronchoscopy regarding the location or nature of foreign bodies, several important factors were not accounted for due to limitations in the available data. Variables such as the size and sharpness of the foreign body, its relative size to the airway lumen, whether the object is fixed or mobile, and more peripheral locations such as subsegmental airways may significantly influence both procedural success and the choice of bronchoscopy method. These factors are crucial and should be considered in clinical decision-making, and future studies should aim to stratify outcomes based on these parameters to provide more nuanced recommendations."
Comment 2. Line 310: The authors noted considerable variability in study populations, types of foreign bodies encountered, and operator expertise, suggesting that the choice between flexible and rigid bronchoscopy might be influenced by these factors rather than inherent differences in efficacy. This acknowledgment raises critical questions about the comparability of the two methods.
Response: Thank you for highlighting this point. We agree that the variability in study populations, types of foreign bodies, and operator expertise raises questions regarding the comparability of flexible and rigid bronchoscopy. These factors may indeed influence outcomes and may overshadow inherent differences between the two techniques. In light of this, we have expanded the discussion to emphasize the limitations this variability imposes on our conclusions, particularly in terms of the generalizability of the findings as follows: "The variability in study populations, types of foreign bodies, and operator expertise observed in the included studies introduces significant challenges in directly comparing the efficacy of flexible and rigid bronchoscopy. These factors likely influence procedural success and safety outcomes, and their impact may obscure any inherent differences between the two techniques. Consequently, the comparability of flexible and rigid bronchoscopy must be interpreted cautiously, as the choice of technique may be influenced by the specific context of each procedure rather than by objective efficacy differences. Future studies should seek to minimize this variability by focusing on more homogeneous populations or by providing stratified analyses that account for operator expertise, foreign body characteristics, and other key variables."
Comment 3. Line 317: The conclusion that recognizing predictors and risk factors can aid in pre-procedural planning, is based on assumptions rather than the results of this study. It largely reiterates findings from previous research.
Response: Thank you for this observation. We agree that our statement regarding predictors and risk factors is more speculative and reflective of prior research rather than being directly supported by the results of our meta-analysis. To clarify, we added the following sentence to that paragraph: "Given the scarcity of data regarding these variables, a quantitative assessment of the impact of these factors in determining examined outcomes was not possible. Therefore, future research should address these points and stratify their outcome data based on these factors."
Comment 4. Line 378: The assertion that both flexible and rigid bronchoscopy are effective and safe for foreign body removal in pediatric patients is questionable due to significant heterogeneity among the studies analyzed, which affects the comparability of the findings.
Response: Thank you for this insightful comment. We acknowledge that the significant heterogeneity observed among the included studies does raise concerns about the direct comparability of flexible and rigid bronchoscopy. This heterogeneity, particularly regarding study design, patient populations, and procedural contexts, limits the strength of the conclusion that both methods are equally effective and safe. We have revised the conclusion to better reflect these limitations and have tempered the generalization of our findings as follows: "While both flexible and rigid bronchoscopy were found to be effective and safe for foreign body removal in pediatric patients based on the available data, the significant heterogeneity observed across the included studies suggests caution in generalizing these findings. The variability in study designs, patient characteristics, and procedural techniques impacts the comparability of the two methods. Therefore, individualized decisions based on the clinical context, patient factors, and operator expertise remain crucial, and further research is needed to refine these conclusions in more homogeneous study settings."
Comment 5. The recommendation that the choice between methods should be guided by specific case characteristics, operator expertise, and the availability of both techniques seems to be based on assumptions, as these factors were not examined in this study and are not derived from the study’s findings.
Response: Thank you for your comment. We have edited the conclusions section accordingly (please check the above comment) and discussed all the limitations in detail in the Limitations and Future Directions section.
Reviewer 3 Report
Comments and Suggestions for Authors
I want to congratulate the authors who underwent a major effort to review and analyze rather vast amount of data in order to come up with important practical conclusions.
1. I find the fact that FB are more likely to be missed with flexible bronchoscope rather odd. It is completely against my experience, and in many ways contra intuitive. However, this is what the studies show. Nevertheless, I think the explanation, which the authors give to this phenomenon cannot be right. There is no way that flexible bronchoscope is inferior to the rigid bronchoscope when it comes to inspection of lower airway, particularly subsegmental bronchi in small children. It has nothing to do with quality of the image, which certainly better with rigid bronchoscope, but rather with size and flexibility of the instrument. I wonder if this phenomenon is related rather to experience of bronchologist, or possibly to the quality of video equipment (E.g monitor vs. visual via the head of the bronchoscope).
I wonder if the authors would agree to make changes to their explanations.
2. I think that the authors make a very appropriate statement related to the value of combined approach with flexible and rigid bronchoscopy to FB removals. This is most likely the way to do it in the ideal situation and it became a "standard of care" in many children's hospital in the US. Did the authors had a chance to review any articles, which describe this very approach? Are there any data to support this statement further?
3. Despite the impressive number of reviewed studies and total amounts of patients included in the studies, I think it is important to recognize that the most of (even larger) pediatric Children's hospitals do not report and do not publish their date on FB removal. That is why "underreporting" is becoming a very significant factor, which may be changing dramatically the results and conclusions. I think that the authors have to acknowledge this fact in the discussion section.
Author Response
Comment 1. I find the fact that FB are more likely to be missed with flexible bronchoscope rather odd. It is completely against my experience, and in many ways contra intuitive. However, this is what the studies show. Nevertheless, I think the explanation, which the authors give to this phenomenon cannot be right. There is no way that flexible bronchoscope is inferior to the rigid bronchoscope when it comes to inspection of lower airway, particularly subsegmental bronchi in small children. It has nothing to do with quality of the image, which certainly better with rigid bronchoscope, but rather with size and flexibility of the instrument. I wonder if this phenomenon is related rather to experience of bronchologist, or possibly to the quality of video equipment (E.g monitor vs. visual via the head of the bronchoscope).
I wonder if the authors would agree to make changes to their explanations.
Response: Thank you for your thoughtful comment and for raising this important issue. We agree that the finding of a higher rate of missed foreign bodies with flexible bronchoscopy is indeed counterintuitive and contrasts with the general perception that flexible bronchoscopy is more suited for inspecting the lower airway, especially in small children. As you mentioned, this may not be due to the inherent limitations of the flexible bronchoscope but rather related to other factors such as the experience of the bronchologist or the quality of the video equipment used. We have revised the manuscript to reflect this possibility and to emphasize that operator experience and equipment quality are crucial factors that warrant further investigation.
We have replaced the previous paragraph with the following paragraph in the Discussion section: "The finding that flexible bronchoscopy is associated with a higher rate of missed foreign bodies, particularly in the lower airways, is indeed unexpected and may not be due to the limitations of the flexible bronchoscope itself. Flexible bronchoscopes are generally better suited for inspecting the subsegmental bronchi, especially in small children, due to their size and flexibility [32]. Conversely, the rigid bronchoscope's larger diameter and direct line of sight might offer superior visualization and mechanical advantage for FB retrieval, especially in proximal airways [33].
It is possible that the higher rate of missed foreign bodies is influenced by the experience of the bronchologist or the quality of the video equipment used (e.g., monitor vs. direct visual inspection). Unfortunately, given the lack of data in this regard, we could not analyze the impact of these factors on this outcome. Further research is needed to explore how these factors impact the effectiveness of flexible bronchoscopy in foreign body detection."
Comment 2. I think that the authors make a very appropriate statement related to the value of combined approach with flexible and rigid bronchoscopy to FB removals. This is most likely the way to do it in the ideal situation and it became a "standard of care" in many children's hospital in the US. Did the authors had a chance to review any articles, which describe this very approach? Are there any data to support this statement further?
Response: Thank you for your positive feedback on our statement regarding the combined use of flexible and rigid bronchoscopy for foreign body removal. We agree that this approach represents an ideal strategy in many settings and is increasingly becoming the "standard of care" in pediatric hospitals, particularly in the United States.
In included studies, the authors tended to report (how many cases underwent both procedures, for whatever reason); however, they did not stratify the baseline and outcome data for this particular patient population. Also, we did not find any studies that particularly included this patient population (who underwent both procedures) at baseline and analyzed their outcomes separately.
Comment 3. Despite the impressive number of reviewed studies and total amounts of patients included in the studies, I think it is important to recognize that the most of (even larger) pediatric Children's hospitals do not report and do not publish their date on FB removal. That is why "underreporting" is becoming a very significant factor, which may be changing dramatically the results and conclusions. I think that the authors have to acknowledge this fact in the discussion section.
Response: Thank you for bringing this important point to our attention. We agree that underreporting, particularly from larger pediatric hospitals, may significantly impact the results and conclusions of meta-analyses like ours. Many institutions may not publish their data on foreign body removal, leading to potential selection bias and underestimation of true procedural outcomes. We have included this point in our discussion as a limitation point as follows: "Despite the impressive number of reviewed studies and the large total number of patients included in our meta-analysis, it is important to recognize the issue of underreporting, particularly from larger pediatric hospitals. Many such institutions may not publish their data on foreign body removal, which could lead to a significant reporting bias. This underreporting may influence the generalizability of our findings and obscure the true rates of success and complications associated with both flexible and rigid bronchoscopy"
Round 2
Reviewer 2 Report
Comments and Suggestions for Authors
Thank you for following up on the reviewer's comments and integrate them into the article.